# Pregnancy outcomes in women with vitiligo: A Taiwanese nationwide cohort study

**Chih-Tsung Hung**[1,2], **Hsin-Hui Huang**[3], **Chun-Kai Wang**[4], **Chi-Hsiang Chung**[5,6], **Chang-Huei Tsao**[7,8], **Wu-Chien Chien**[5,7,9]*, **Wei-Ming Wang**[1,2]*

1 Department of Dermatology, National Defense Medical Center, Tri-Service General Hospital, Taipei, Taiwan, 2 National Defense Medical Center, Graduate Institute of Medical Sciences, Taipei, Taiwan, 3 Department of Obstetrics and Gynecology, National Defense Medical Center, Tri-Service General Hospital, Taipei, Taiwan, 4 Department of Obstetrics and Gynecology, Zuoying Branch of Kaohsiung Armed Forces General Hospital, Kaohsiung, Taiwan, 5 School of Public Health, National Defense Medical Center, Taipei, Taiwan, 6 Taiwanese Injury Prevention and Safety Promotion Association, Taipei, Taiwan, 7 Department of Medical Research, National Defense Medical Center, Tri-Service General Hospital, Taipei, Taiwan, 8 Department of Microbiology & Immunology, National Defense Medical Center, Taipei, Taiwan, 9 Graduate Institute of Life Sciences, National Defense Medical Center, Taipei, Taiwan

* chienwu@ndmctsgh.edu.tw (WC); ades0431@ms38.hinet.net (WW)

**Data Availability Statement:** All relevant data are within the manuscript.

**Funding:** This study was supported by grants from Tri-Service Hospital Research Foundation (TSGH-

## Abstract

Vitiligo is perceived as an autoimmune skin disease. Previous studies showed conflicting data about vitiligo and pregnancy outcomes. To delineate the associations between vitiligo and the pregnancy outcomes, we used the National Health Insurance Research Database of Taiwan to conduct a retrospective cohort study from January 1, 2000 to December 31, 2015. This study population was composed of 1,096 women with vitiligo and 4,384 women without vitiligo, who were all matched according to age, comorbidity, and index year. Compared with the non-vitiligo controls, women with vitiligo had a higher risk of abortion (aHR 1.158, 95% confidence interval (CI) 1.095–1.258, P < .001). Perinatal events, such as preterm delivery, pre-eclampsia/eclampsia, gestational diabetes mellitus, stillbirth, and intrauterine growth retardation, were not different between both groups (aHR 1.065, 95% CI 0.817–1.157, P = .413). To determine if systemic treatment before conception decreases the risk of abortion, we assessed the medical history of pregnant women with vitiligo 1 year before pregnancy. Patients who were treated with oral medications had a lower risk of abortion than those who were not (aHR: 0.675, 95% CI: 0.482–0.809, P < .001). Our study indicates that there is a higher risk of abortion in pregnant women with vitiligo and the control of disease activity with systemic treatment before conception could improve pregnancy outcomes.

## Introduction

Vitiligo is an acquired skin disease with progressive dermal and mucosal depigmentation. In this condition, CD8+ cytotoxic T cells destruct the epidermal melanocytes [1, 2]. Depending on the different studied populations, its prevalence varies from 0.064% to 1% [3, 4]. Patients with vitiligo may feel stigmatized, have poor body image, have low self-esteem, and experience

B-109-010, TSGH-D-109-050, and TSGH-E110219). The funders had no role in study design, data collection and analysis, decision to publish, or preparation of the manuscript.

**Competing interests:** The authors have declared that no competing interests exist.

extreme psychological distress [5, 6]. A recent meta-analysis study revealed that one in four people with vitiligo suffers from depression [7].

Vitiligo is not only a cosmetic problem but also a chronic autoimmune disease. Several systemic autoimmune diseases, such as systemic lupus erythematosus (SLE), Sjögren's syndrome, rheumatoid arthritis (RA), and autoimmune thyroiditis (e.g., Hashimoto thyroiditis and Graves' disease) are associated with vitiligo [3, 8]. Autoimmune diseases are also associated with obstetric complications [9, 10]. However, the association between vitiligo and pregnancy outcomes remains unclear [11, 12]. Therefore, this study aimed to delineate the association between vitiligo and pregnancy outcomes such as abortion, preterm delivery, pre-eclampsia/eclampsia, gestational diabetes mellitus (DM), stillbirths, intrauterine growth retardation (IUGR) by using Taiwan National Health Insurance Research Database.

## Material and methods

### Ethics statement

Our study conformed to the principles of the Declaration of Helsinki and was approved by the Institutional Review Board of Tri-Service General Hospital (TSGHIRB C202005017). Considering that the Taiwan National Health Insurance Research Database (NHIRD) contains fully anonymized linked information, which does not affect the welfare and right of the subjects, the requirement for informed consent from our participants was waived.

### Data sources

In Taiwan, >99% of the population has joined the National Health Insurance, which is a mandatory single-payer social insurance. The Taiwan NHIRD contains many anonymized linked data, including outpatient care, inpatient care, emergency care, treatment procedures, and prescription drugs. Physicians diagnosed their patients using the International Classification of Diseases, 9th Revision, Clinical Modification (ICD-9-CM) in the NHIRD between 2000 and 2015. The diagnoses in the NHIRD have been proven accurate by numerous previous studies [13, 14].

### Study participants

The selection of the study participants is depicted in Fig 1. We enrolled patients with vitiligo (ICD-9-CM code 709.01) and with data of their first pregnancy (ICD-9-CM code V22-V23) from the Longitudinal Health Insurance Database between January 1, 2000 and December 31, 2015 in Taiwan. We evaluated only the first pregnancy to avoid bias related to multiple outcomes in the same women. Meanwhile, we excluded those who were pregnant or diagnosed with SLE, Sjögren's syndrome, or RA before 2000, or before the first visit for vitiligo. All patients aged below 20 years were also excluded. The study population was composed of 1,096 women with vitiligo and 4,384 women without vitiligo, who were all matched according to age, comorbidity, and index year.

### Outcome measures

Pregnancy outcomes included live births (ICD-9-CM code 650–655, 656.0–656.3, 656.6–656.9, 669.50–669.51), abortion (ICD-9-CM code 632, 634, 637), cesarean delivery (ICD-9-CM code 669.7), and other perinatal events. Elective abortion (ICD-9-CM code 779.6) was also excluded in both groups. Perinatal events comprised preterm delivery (ICD-9-CM code 644.0–64.2), pre-eclampsia/eclampsia (ICD-9-CM code 642.4–642.7), gestational DM (ICD-9-CM code 648.8), stillbirths (ICD-9-CM code 656.4), and IUGR (ICD-9-CM code 656.5).

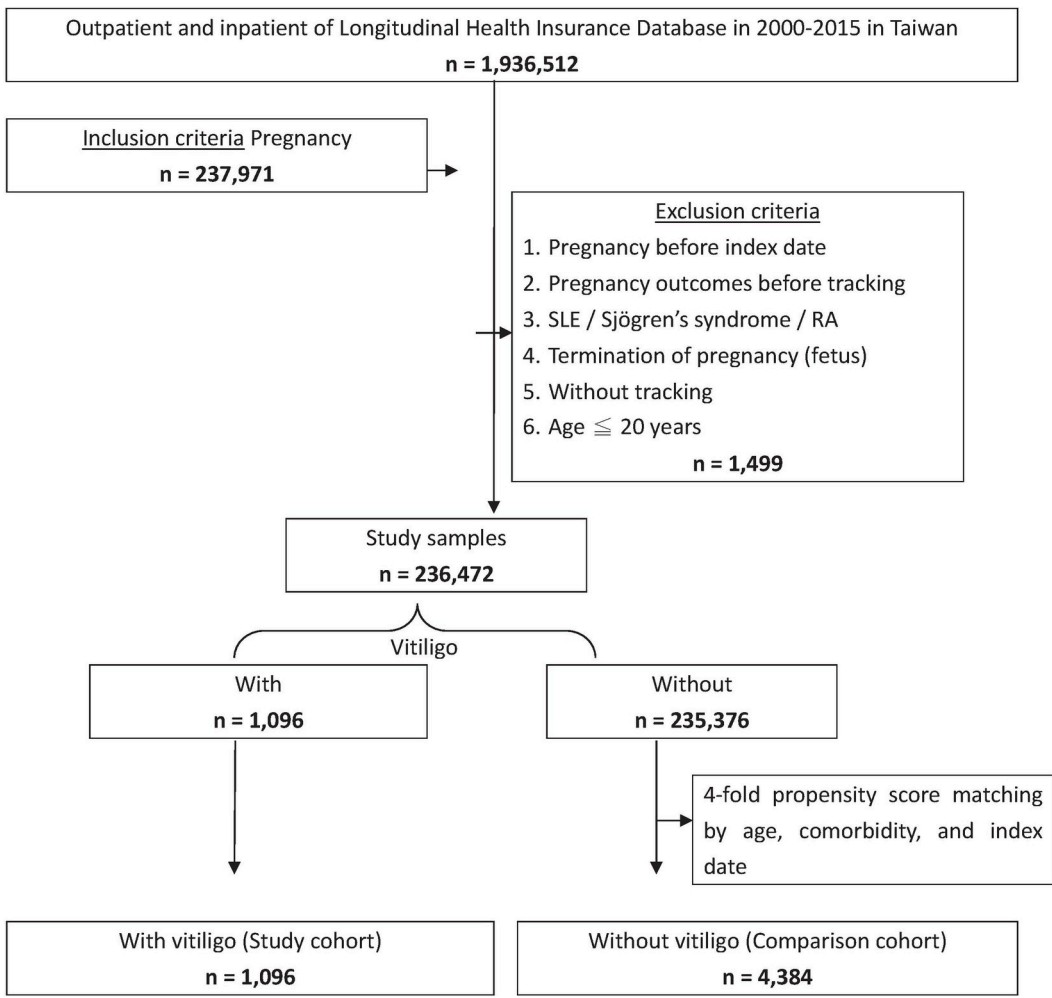

**Fig 1. The flowchart of study population selection.** This study included 1,096 first and subsequent singleton pregnancies in women with vitiligo and 4,384 pregnancies in women without vitiligo from 2000 to 2015.

Until the end of the study, we assessed the comorbid diseases of women who had experienced abortion. These diseases included SLE (ICD-9-CM code 710.0), Sjögren's syndrome (ICD-9-CM code 710.2), RA (ICD-9-CM code 714.0), Graves' disease (ICD-9-CM code 242), Hashimoto thyroiditis (ICD-9-CM code 245.2), psoriasis (ICD-9-CM code 696.1), and atopic dermatitis (ICD-9-CM code 691.8).

Furthermore, we evaluated whether systemic treatment before conception decreases the risk of abortion and assessed the medical history of women with vitiligo 1 year before pregnancy. We determined the prescribed oral medications and phototherapy by using drug and procedure number in the NHIRD.

## Covariates

The covariates included the age groups (20–29, 30–39, and 40–49 years), comorbidity, seasons, level of care, geographical area of residence, and urbanization level. To evaluate comorbidities, we used Charlson comorbidity index (CCI), which comprises various comorbidities, including cerebrovascular disease, dementia, hemiplegia, chronic pulmonary disease, congestive heart failure, myocardial infarction, peripheral vascular disease, connective tissue disease, diabetes,

**Table 1. Maternal characteristics of study population in the baseline.**

| Vitiligo | Total | | With | | Without | | P |
|---|---|---|---|---|---|---|---|
| Variables | N | % | n | % | N | % | |
| Total | 5,480 | | 1,096 | 20.00 | 4,384 | 80.00 | |
| Age (years) | 29.53±5.31 | | 29.62±5.48 | | 29.51±5.27 | | 0.540 |
| Age groups (years) | | | | | | | 0.999 |
| 20–29 | 2,865 | 52.28 | 573 | 52.28 | 2,292 | 52.28 | |
| 30–39 | 2,560 | 46.72 | 512 | 46.72 | 2,048 | 46.72 | |
| 40–49 | 55 | 1.00 | 11 | 1.00 | 44 | 1.00 | |
| CCI | 1.04±1.94 | | 1.10±1.85 | | 1.03±1.96 | | 0.285 |

P: Chi-squared/Fisher's exact test on category variables and t-test on continue variables. CCI, Charlson comorbidity index.

renal and liver disease, ulcer disease, any tumor, leukemia, lymphoma, and AIDS; all comorbidity scores in each study group were totaled to obtain the CCI [15]. The higher the score, the more the comorbidity burdens.

## Statistical analysis

Categorical variables were compared using the chi-squared test, whereas the continuous variables were evaluated using Fisher' exact test and *t*-test. We calculated the incidence rate (per 105 person-months) by dividing the number of women with perinatal events by the total person-months of each group. Adjusted hazard ratios and 95% confidence intervals (CIs) were calculated using the Multivariable Cox proportional hazard model. A two-tailed *P* value < .001 and a CI not including 1 were considered statistically significant. All statistical data were analyzed using the SPSS software version 22 (SPSS Inc., Chicago, Ill., USA).

## Results

This cohort study included 1,096 pregnant women with vitiligo and 4,384 pregnant women without vitiligo who were both recorded in the NHIRD between 2000 and 2015. Both study groups had similar age and comorbidities (Table 1). The peak age group of pregnancy was 20–29 years (52.28%), followed by 30–39 years (46.72%).

After adjusting for age, comorbidity, season, location, urbanization level, and level of care, the women with vitiligo had a lower live-birth rate (aHR: 0.800, 95% CI: 0.727–0.824, *P* < .001, Table 2) and a higher abortion rate (aHR: 1.158, 95% CI: 1.095–1.258, *P* < .001) than those without vitiligo. Subgroup analysis of the total perinatal events, including preterm delivery, pre-eclampsia/eclampsia, gestational DM, stillbirths, and IUGR, revealed no significant differences between both groups (aHR: 1.065, 95% CI: 0.817–1.157, *P* = .413).

To determine if systemic treatment before conception decreases risk of abortion, we assessed the medical history of pregnant women with vitiligo 1 year before the pregnancy; those who received oral medications and/or phototherapy for at least 3 months were included in the systemic treatment group. Patients with vitiligo undergoing systemic treatment (*n* = 296) were compared with those without systemic treatment (*n* = 800, Table 3). Patients who were treated with oral medications had a lower risk of abortion than those who were not (aHR: 0.675, 95% CI: 0.482–0.809, *P* < .001). In the subgroup analysis of patients with oral medications, the risk of abortion was lowest in patients taking oral steroids (aHR: 0.596, 95% CI: 0.463–0.774, *P* < .001), followed by those taking methotrexate (aHR: 0.720, 95% CI: 0.575–0.873, *P* < .001). Meanwhile, patients treated with azathioprine demonstrated no significant

**Table 2. Risks of maternal and infant outcomes in women with vitiligo compared with those without vitiligo.**

| Vitiligo | With n = 1,096 | | | Without n = 4,384 | | | With vs. Without (Reference) | | | |
|---|---|---|---|---|---|---|---|---|---|---|
| Pregnancy outcomes | Events | PMs | Rate (per $10^5$ PMs) | Events | PMs | Rate (per $10^5$ PMs) | Adjusted HR | 95% CI | 95% CI | P |
| Live births | 634 | 13,960.70 | 4,541.32 | 2,679 | 55,841.46 | 4,797.51 | 0.800 | 0.727 | 0.824 | <0.001 |
| Abortion | 117 | 13,823.46 | 846.39 | 354 | 55,308.49 | 640.05 | 1.158 | 1.095 | 1.258 | <0.001 |
| Cesarean delivery | 260 | 13,816.85 | 1,881.76 | 1,047 | 55,306.39 | 1,893.09 | 0.901 | 0.797 | 1.008 | 0.425 |
| **Perinatal events** | | | | | | | | | | |
| Total | 81 | 13,923.60 | 581.75 | 304 | 55,643.17 | 546.34 | 1.065 | 0.817 | 1.157 | 0.413 |
| Preterm delivery | 24 | 12,963.62 | 185.13 | 86 | 54,333.79 | 158.28 | 1.028 | 0.788 | 1.168 | 0.362 |
| Pre-eclampsia / eclampsia | 21 | 13,487.39 | 155.70 | 71 | 54,525.03 | 130.22 | 1.157 | 0.783 | 1.258 | 0.264 |
| Gestational DM | 14 | 13,923.56 | 100.55 | 58 | 55,643.17 | 104.24 | 1.012 | 0.715 | 1.108 | 0.699 |
| Stillbirths | 9 | 13,892.33 | 64.78 | 35 | 55,538.33 | 63.02 | 1.004 | 0.651 | 1.091 | 0.734 |
| IUGR | 13 | 13,894.55 | 93.56 | 54 | 55,587.60 | 97.14 | 1.031 | 0.763 | 1.334 | 0.415 |

Adjusted HR, Adjusted Hazard ratio: adjusted with age, comorbidity, season, location, urbanization level, and level of care; CI = confidence interval; DM, diabetes mellitus; IUGR, intrauterine growth retardation; PMs = Person-months

differences (aHR: 0.837, 95% CI: 0.746–1.238, P = .211). Moreover, seven patients underwent combined oral medications and phototherapy, and no abortion was observed.

To identify the incidence of various comorbid diseases, we followed women who experienced abortion until the end of study (Table 4). Patients with vitiligo who experienced abortion had a higher risk of SLE (aHR: 2.146, 95% CI: 1.325–2.897, P < .001), followed by the risks of RA (aHR: 2.068, 95% CI 1.297–2.568, P < .001) and Sjögren's syndrome (aHR: 1.873, 95% CI: 1.124–2.335, P < .001), than those without vitiligo. Furthermore, Hashimoto thyroiditis was identified in one patient in the vitiligo group, but none in the control group. Meanwhile, psoriasis and atopic dermatitis revealed no significant differences.

## Discussion

Currently, data about the association between vitiligo and pregnancy outcomes are inconsistent [11, 12]. In a single-institution, retrospective, and comparative study, the pregnancy outcomes of 79 women with vitiligo and 186,143 controls without vitiligo were not different [11].

**Table 3. Systemic treatment before conception reduced risk of abortion in women with vitiligo.**

| Pregnancy outcomes | Vitiligo | Population | Outcomes | PMs | Rate (per $10^5$ PMs) | Adjusted HR | 95% CI | 95% CI | P |
|---|---|---|---|---|---|---|---|---|---|
| **Abortion** | **Total** | 1,096 | 117 | 13,823.46 | 846.39 | | | | |
| | **Without systemic treatment** | 800 | 95 | 10,446.40 | 909.40 | Reference | | | |
| | **Oral medication or Phototherapy** | 296 | 22 | 3,377.06 | 651.45 | 0.755 | 0.710 | 0.979 | 0.030 |
| | **Oral medication only** | 231 | 17 | 2,554.19 | 665.57 | 0.675 | 0.482 | 0.809 | <0.001 |
| | oral steroid only | 171 | 12 | 1,915.51 | 626.46 | 0.596 | 0.463 | 0.774 | <0.001 |
| | oral MTX only | 36 | 3 | 406.18 | 738.59 | 0.720 | 0.575 | 0.873 | <0.001 |
| | oral AZA only | 24 | 2 | 232.50 | 860.22 | 0.837 | 0.746 | 1.238 | 0.211 |
| | **Phototherapy only** | 58 | 5 | 622.09 | 803.74 | 0.821 | 0.738 | 0.987 | 0.034 |
| | **Combined oral medication and phototherapy** | 7 | 0 | 200.78 | 0.00 | 0.000 | - | - | 0.975 |

Adjusted HR, Adjusted Hazard ratio: adjusted with age, comorbidity, season, location, urbanization level, and level of care; CI = confidence interval; PMs, Person-months; AZA, Azathioprine; MTX, methotrexate.

**Table 4. Rates of various comorbid diseases among women with abortion.**

| Vitiligo | With (n = 117) | | | Without (n = 354) | | | With *vs.* Without *(Reference)* | | | |
|---|---|---|---|---|---|---|---|---|---|---|
| Comorbid diseases | Events | PYs | Rate (per $10^5$ PYs) | Events | PYs | Rate (per $10^5$ PYs) | Adjusted HR | 95% CI | 95% CI | *P* |
| SLE | 4 | 1,593.16 | 251.07 | 6 | 5,434.06 | 110.41 | 2.146 | 1.325 | 2.897 | <0.001 |
| RA | 6 | 1,541.18 | 389.31 | 10 | 5,306.10 | 188.46 | 2.068 | 1.297 | 2.568 | <0.001 |
| Sjögren's syndrome | 10 | 1,478.26 | 676.47 | 19 | 5,264.65 | 360.90 | 1.873 | 1.124 | 2.335 | <0.001 |
| Graves' disease | 0 | 1,611.87 | 0.00 | 0 | 5,574.69 | 0.00 | - | - | - | - |
| Hashimoto thyroiditis | 1 | 1,612.51 | 62.02 | 0 | 5,547.25 | 0.00 | - | - | - | - |
| Psoriasis | 2 | 1,584.28 | 126.24 | 6 | 5,462.76 | 109.83 | 1.125 | 0.523 | 1.735 | 0.437 |
| Atopic dermatitis | 2 | 1,608.26 | 124.36 | 4 | 5,490.65 | 72.85 | 1.402 | 0.861 | 2.001 | 0.288 |

Adjusted HR, Adjusted Hazard ratio: adjusted with age, comorbidity, season, location, urbanization level, and level of care; CI = confidence interval; PYs, Person-years; RA, Rheumatoid arthritis; SLE, Systemic lupus erythematosus

In 2018, a large Korean cohort study using the Korean National Health Insurance Claims database from 2007 to 2016 was published [12]. The results of this previous study revealed a significant lower rate of live births (odds ratio: 0.780, *P* < .001) and a higher rate of spontaneous abortion (odds ratio: 1.250, *P* < .001) in pregnant women with vitiligo (*n* = 4,738) than in pregnant age-matched controls without vitiligo (*n* = 47,380). In addition, a more extensive disease was associated with a higher risk of abortion and a lower live-birth rate than limited disease. In this study, we revealed that pregnant women with vitiligo had a higher risk of abortion, with an aHR of 1.158 (*P* < .001), and a lower live-birth rate, with an aHR of 0.800 (*P* < .001).

Vitiligo is a kind of autoimmune disease in which T cells are involved in the pathogenesis [16–18]. In a recent study, the number of circulating Th1 and Th17 were higher in patients with vitiligo than in healthy controls. In active nonsegmental vitiligo, the ratios of Th1/Tregs and Th17/Tregs in circulation are both elevated [19]. The Th1/Th2 balance that switches toward a Th2-dominated cytokine profile during pregnancy might be an important mechanism for maternal immunotolerance of the fetus [20, 21]. However, Th1 immune response may be associated with abortion [22]. A previous study had characterized maternal CD4 T-cell effector subsets during pregnancy (women with normal pregnancy, cross-sectional cohort, *n* = 71; longitudinal cohort, *n* = 17; compared with women with recurrent miscarriage, *n* = 24) by longitudinal analysis of the peripheral blood, and the results revealed that women with recurrent miscarriage had an increased proportion of Th1 and Th17 cells and related cytokines such as IFN-γ and IL-17 [23]. Therefore, immune dysregulation in vitiligo may cause abortion and reduce the rate of live births.

Flares of autoimmune disease activity during pregnancy are unfavorable, increasing the risk of abortion and premature birth [24, 25]. In our study, we sought to understand the impact of systemic treatment on pregnancy outcomes. We found that the patients with vitiligo treated with oral medication before the pregnancy had a decreased risk of abortion. Therefore, disease control with systemic treatment before conception seems to be important, but further investigation is required to prove these findings.

Moreover, patients with vitiligo have higher risks of autoimmune diseases, thyroiditis, alopecia areata, atopic dermatitis, and psoriasis [3, 8, 26–28]. In Taiwan, a large nationwide study presented by Chen et al. revealed that the risks of SLE, Sjögren's syndrome, and RA were higher in patients with vitiligo aged 60–79 years [3]. In this study, we followed the women (aged 20–49 years) who had experienced abortion until the end of the study to determine the various comorbid diseases. The results showed a higher risk of SLE, followed by RA and

Sjögren's syndrome, in patients with vitiligo who had experienced abortion than in controls without vitiligo. Regarding thyroiditis, one case of Hashimoto thyroiditis was found in the vitiligo group but none in the control group.

As a strength of this study, it has a nationwide retrospective cohort study design utilized for investigating pregnancy outcomes in women with vitiligo in Taiwan. By using NHIRD, we not only could trace back the medical history before pregnancy to determine the impact of systemic treatment on the pregnancy outcomes but also identify the comorbid diseases among women who manifested abortion throughout the study period. However, this study also has some limitations. First, the NHIRD data lack clinical presentations of vitiligo, including body surface area, lesion sites, family history, laboratory parameters, genetics, alcohol use, smoking, and the body mass index. Therefore, we cannot elucidate whether these confounders and the lesion size affect the associations between vitiligo and pregnancy outcomes. Second, the main study population was Taiwanese. Thus, ethnic discrepancy could exist. Third, coding error is possible. To minimize this bias, we defined study patients with vitiligo as those who were diagnosed by dermatologists, and the pregnancy outcomes were coded by obstetricians.

## Conclusions

Pregnant women with vitiligo were significantly associated with an increased risk of abortion, and systemic treatment before conception could improve pregnancy outcomes.

## Acknowledgments

This study is based in part on data form the National Health Insurance Research Database provided by the Health and Welfare Data Science center, Ministry of Health and Welfare (HWDC, MOHW), and managed by National Health Research Institutes.

## Author Contributions

**Conceptualization:** Chih-Tsung Hung.

**Data curation:** Chih-Tsung Hung, Wu-Chien Chien.

**Formal analysis:** Chi-Hsiang Chung, Chang-Huei Tsao, Wu-Chien Chien.

**Funding acquisition:** Chih-Tsung Hung, Wei-Ming Wang.

**Investigation:** Chi-Hsiang Chung.

**Methodology:** Chih-Tsung Hung, Hsin-Hui Huang, Chun-Kai Wang, Chi-Hsiang Chung, Chang-Huei Tsao, Wu-Chien Chien.

**Resources:** Wu-Chien Chien.

**Software:** Wu-Chien Chien.

**Supervision:** Wu-Chien Chien, Wei-Ming Wang.

**Validation:** Wu-Chien Chien, Wei-Ming Wang.

**Writing – original draft:** Chih-Tsung Hung.

**Writing – review & editing:** Wu-Chien Chien, Wei-Ming Wang.

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
