## [Decision Letter · Decision Letter 0]

3 Feb 2021

PONE-D-20-40830

Pregnancy outcomes in women with vitiligo: a Taiwan nationwide cohort study

PLOS ONE

Dear Dr. Hung,

Thank you for submitting your manuscript to PLOS ONE. After careful consideration, we feel that it has merit but does not fully meet PLOS ONE’s publication criteria as it currently stands. Therefore, we invite you to submit a revised version of the manuscript that addresses the points raised during the review process.

We look forward to receiving your revised manuscript.

Kind regards,

Christy Pu

Academic Editor

PLOS ONE

Journal Requirements:

2) In the ethics statement in the manuscript and in the online submission form, please provide additional information about the patient records used in your retrospective study, including: a) whether all data were fully anonymized before you accessed them; b) the date range (month and year) during which patients' medical records were accessed. If patients provided informed written consent to have data from their medical records used in research, please include this information.

3) Your ethics statement should only appear in the Methods section of your manuscript. If your ethics statement is written in any section besides the Methods, please delete it from any other section.

4)  We suggest you thoroughly copyedit your manuscript for language usage, spelling, and grammar. If you do not know anyone who can help you do this, you may wish to consider employing a professional scientific editing service.  

5)  Thank you for stating the following in the Acknowledgments/Funding sources Section of your manuscript:

[This study was supported by grants from Tri-Service Hospital Research

Foundation (TSGH-B-109-010, TSGH-D-109-050, and 801GB110220).]

 [The funders had no role in study design, data collection and analysis, decision to

publish, or preparation of the manuscript.]

Reviewers' comments:

Reviewer's Responses to Questions

5. Review Comments to the Author

Reviewer #1: 1. The study of the relations between pregnant outcomes and vitiligo is quite rare, this research points out some unknown features by applying large scale research database.

2. The Taiwan National Health Insurance (NHI) and the National Health Insurance Research Database (NHIRD) should be different. Are there any references indicating the coverage rate (>99% in the text) and the reimbursement system of NHI? Further, how it works between the NHI and NHIRD? Third, the abbreviation of NHIRD should be used when it first appeared in the manuscript.

3. As mentioned in the study limitation section, the extend and severity of the vitiligo is difficult to be defined, therefore, the treatment of choice remained controversy. Topical agents have longstanding been used as the first line therapy, why only systemic treatments were regarded as preconception treatment? Consequently, should the 800 cases who were assigned as “without treatment” be “real non treatment” or “topical treatment”? Since those cases without treatment also remained controversy, the term used in the table 3 should be revised, and the conclusion also should be reconsidered accordingly.

4. As for the comorbid diseases, the SLE/Sjogren’s syndrome/RA were excluded for the enrollment in the beginning, so how to explain why those diseases still existed in the abortion group of vitiligo patients? Furthermore, immunosuppressive agents such as methotrexate and azathioprine are not common as the systemic medications for vitiligo prescribed by dermatologists, is there any explanation?

---

## [Author Response · Author response to Decision Letter 0]

15 Feb 2021

1. The study of the relations between pregnant outcomes and vitiligo is quite rare, this research points out some unknown features by applying large scale research database.

Response:

Thanks for your kind comment. The purpose of our study is to emphasize the importance of vitiligo in pregnant women. Out of personal discussion, most obstetricians would check whether pregnant patients were previously diagnosed as SLE or Sjögren’s syndrome, but vitiligo which is also an autoimmune disease seldom drew their attention. Besides, our study also revealed the importance of disease control with systemic treatment to reduce the risk of miscarriage. 

2. The Taiwan National Health Insurance (NHI) and the National Health Insurance Research Database (NHIRD) should be different. Are there any references indicating the coverage rate (>99% in the text) and the reimbursement system of NHI? Further, how it works between the NHI and NHIRD? Third, the abbreviation of NHIRD should be used when it first appeared in the manuscript.

Response:

Thanks for your kind comment and reminder. Taiwan National Health Insurance (NHI) is a single-payer compulsory social insurance. This system promises equal access to healthcare for all citizens, and the coverage had reached 99% [1]. NHIRD, the large computerized databases, derived from the National Health Insurance Administration (the former Bureau of National Health Insurance, BNHI), Ministry of Health and Welfare (the former Department of Health, DOH), Taiwan and maintained by the National Health Research Institutes, Taiwan, are provided to scientists in Taiwan for research purposes. The data in NHIRD is fully anonymized linked information, including outpatient care, inpatient care, emergency care, treatment procedures, and prescription drugs. 

3. As mentioned in the study limitation section, the extend and severity of the vitiligo is difficult to be defined, therefore, the treatment of choice remained controversy. Topical agents have longstanding been used as the first line therapy, why only systemic treatments were regarded as preconception treatment? Consequently, should the 800 cases who were assigned as “without treatment” be “real non treatment” or “topical treatment”? Since those cases without treatment also remained controversy, the term used in the table 3 should be revised, and the conclusion also should be reconsidered accordingly.

Response:

Thanks for your kind reminder. As your comment, topical agents have longstanding been used as the first line therapy. In this study, we want to know if the systemic treatment before pregnancy decreases the risk of abortion or not, and we revised our article accordingly, including manuscript and tables. 

4. As for the comorbid diseases, the SLE/Sjogren’s syndrome/RA were excluded for the enrollment in the beginning, so how to explain why those diseases still existed in the abortion group of vitiligo patients? Furthermore, immunosuppressive agents such as methotrexate and azathioprine are not common as the systemic medications for vitiligo prescribed by dermatologists, is there any explanation?

Response:

Thanks for your kind comment. This is a retrospective cohort study ranging from January 1st, 2000 to December 31st, 2015. SLE/Sjogren’s syndrome/RA were excluded for the enrollment in the beginning, and we rechecked our enrolled data that no one was diagnosed of SLE/Sjogren’s syndrome/RA before their first pregnancy. We kept following the medical records after their first pregnancy and found the increased risk of SLE/Sjogren’s syndrome/RA in patients with vitiligo who experienced abortion. Besides, steroid phobia is a common problem in my country. Some patients rejected any treatment containing steroid, including topical and oral agents. At that time, we will arrange phototherapy and/or prescribe topical calcineurin inhibitors and/or oral immunosuppressive agents, such as methotrexate and azathioprine, to control their vitiligo disease activity. 

Reference

1. Wu T-Y, Majeed A, Kuo KN. An overview of the healthcare system in Taiwan. London journal of primary care. 2010;3(2):115-9.

---

## [Editor Report · Decision Letter 1]

23 Feb 2021

PONE-D-20-40830R1

Pregnancy outcomes in women with vitiligo: a Taiwanese nationwide cohort study

PLOS ONE

Dear Dr. Hung,

Thank you for submitting your manuscript to PLOS ONE. After careful consideration, we feel that it has merit but does not fully meet PLOS ONE’s publication criteria as it currently stands. Therefore, we invite you to submit a revised version of the manuscript that addresses the points raised during the review process.

We look forward to receiving your revised manuscript.

Kind regards,

Christy Pu

Academic Editor

PLOS ONE

Journal Requirements:

Additional Editor Comments:

1. Please remove results from Discussion.

2. Odds ratios among different studies (or models) should never be compared in terms of magnitudes. This is a common mistake. Please remove all comparisons and rephrase all relevant discussions.

3. In the Limitation section, please discuss briefly direction of bias and how they may affect your conclusion.

---

## [Author Response · Author response to Decision Letter 1]

26 Feb 2021

1. Please remove results from Discussion.

Response:

Thanks for your kind comment. The results were removed from discussion in the revised article accordingly.

2. Odds ratios among different studies (or models) should never be compared in terms of magnitudes. This is a common mistake. Please remove all comparisons and rephrase all relevant discussions.

Response:

Thanks for your kind comment and reminder. We revised and rephrased our article accordingly.

3. In the Limitation section, please discuss briefly direction of bias and how they may affect your conclusion.

Response:

Thanks for your kind comment and reminder. We took your recommendation and revised our article accordingly.

---

## [Editor Report · Decision Letter 2]

3 Mar 2021

Pregnancy outcomes in women with vitiligo: a Taiwanese nationwide cohort study

PONE-D-20-40830R2

Dear Dr. Hung,

We’re pleased to inform you that your manuscript has been judged scientifically suitable for publication and will be formally accepted for publication once it meets all outstanding technical requirements.

Kind regards,

Christy Pu

Academic Editor

PLOS ONE

---

## [Editor Report · Acceptance letter]

12 Mar 2021

PONE-D-20-40830R2 

Pregnancy outcomes in women with vitiligo: a Taiwanese nationwide cohort study 

Dear Dr. Wang:

I'm pleased to inform you that your manuscript has been deemed suitable for publication in PLOS ONE. Congratulations! Your manuscript is now with our production department. 

Kind regards, 

on behalf of

Dr. Christy Pu 

Academic Editor

PLOS ONE